# Policy Optimization in Zero-Sum Markov Games: Fictitious Self-Play Provably Attains Nash Equilibria

## Abstract

Fictitious Self-Play (FSP) has achieved significant empirical success in solving extensive-form games. However, from a theoretical perspective, it remains unknown whether FSP is guaranteed to converge to Nash equilibria in Markov games. As an initial attempt, we propose an FSP algorithm for two-player zero-sum Markov games, dubbed as smooth FSP, where both agents adopt an entropy-regularized policy optimization method against each other. Smooth FSP builds upon a connection between smooth fictitious play and the policy optimization framework. Specifically, in each iteration, each player infers the policy of the opponent implicitly via policy evaluation and improves its current policy by taking the smoothed best-response via a proximal policy optimization (PPO) step. Moreover, to tame the non-stationarity caused by the opponent, we propose to incorporate entropy regularization in PPO for algorithmic stability. When both players adopt smooth FSP simultaneously, i.e., with self-play, in a class of games with Lipschitz continuous transition and reward, we prove that the sequence of joint policies converges to a neighborhood of a Nash equilibrium at a sublinear $\widetilde{O}(1/T)$ rate, where $T$ is the number of iterations. To our best knowledge, we establish the first finite-time convergence guarantee for FSP-type algorithms in zero-sum Markov games.

## 1 Introduction

Multi-agent reinforcement learning (MARL) (Bu et al., 2008; Sutton & Barto, 2018) has achieved great empirical success, e.g., in playing the game of Go (Silver et al., 2016; 2017), Dota 2 (Berner et al., 2019), and StarCraft 2 (Vinyals et al., 2019), which are all driven by policy optimization algorithms which iteratively update the policies that are parameterized using deep neural networks. Empirically, the popularity of policy optimization algorithms for MARL is attributed to the observations that they usually converges faster than value-based methods that iteratively update the value functions (Mnih et al., 2016; O'Donoghue et al., 2016).

Compared with their empirical success, the theoretical aspect of policy optimization algorithms in MARL setting (Littman, 1994; Hu & Wellman, 2003; Conitzer & Sandholm, 2007; Pérolat et al., 2016; Zhang et al., 2018) remains less understood. Although convergence guarantees for various policy optimization algorithms have been established under the single-agent RL setting (Sutton et al., 2000; Konda & Tsitsiklis, 2000; Kakade, 2002; Agarwal et al., 2019; Wang et al., 2019), extending those theoretical guarantees to arguably one of the simplest settings of MARL, two-player zero-sum Markov game, suffers from challenges in the following two aspects. First, in such a Markov game, each agent interact with the opponent as well as the environment. Seen from the perspective of each agent, it belongs to an environment that is altered by the actions of the opponent. As a result, due to the existence of an opponent, the policy optimization problem of each agent has a time-varying objective function, which is in stark contrast with the value-based methods such as value-iteration Shapley (1953); Littman (1994), where there is a central controller which specifies the policies of both players. When the joint policy of both players are considered, the problem of solving the optimal value function corresponds to finding the fixed point of the Bellman operator, which is defined independently of the policy of the players. Second, when viewing the policy optimization in zero-sum Markov game as an optimization problem for both players together, although we have

a fixed objective function, the problem is minimax optimization with a non-convex non-concave objective. Even for classical optimization, such a kind of optimization problem remains less less understood (Cherukuri et al., 2017; Rafique et al., 2018; Daskalakis & Panageas, 2018; Mertikopoulos et al., 2018). It is observed that first-order methods such as gradient descent might fail to converge (Balduzzi et al., 2018; Mazumdar & Ratliff, 2018).

As an initial step to study policy optimization for MARL, we propose a novel policy optimization algorithm for any player of a multi-player Markov game, which is dubbed as smooth fictitious self-play (FSP). Specifically, when a player adopts smooth FSP, in each iteration, it first solves a policy evaluation problem that estimates the value function associate with the current joint policy of all players. Then it update its own policy via an entropy-regularized proximal policy optimization (PPO) Schulman et al. (2017) step, where the update direction is obtained from the estimated value function. This algorithm can be viewed as an extension of the fictitious play (FP) algorithm that is designed for normal-form games (Von Neumann & Morgenstern, 2007; Shapley, 1953) and extensive-form games (Heinrich et al., 2015; Perolat et al., 2018) to Markov-games. FP is a general algorithmic framework for solving games where an agent first infer the policy of the opponents and then adopt a policy that best respond to the inferred opponents. When viewing our algorithm as a FP method, instead of estimating the policies of the opponents directly, the agent infers the opponent implicitly by estimating the value function. Besides, policy update corresponds to a smoothed best-response policy Swenson & Poor (2019) based on the inferred value function.

To examine the theoretical merits of the proposed algorithm, we focus on two-player zero-sum Markov games and let both players follow smooth FSP, i.e., with self-play. Moreover, we restrict to a class of Lipschitz games (Radanovic et al., 2019) where the impact of each player's policy change on the environment is Lipschitz continuous with respect to the magnitude of policy change. For such a Markov game, we tackle the challenge of non-stationarity by imposing entropy regularization which brings algorithmic stability. In addition, to establish convergence to Nash equilibrium, we explicitly characterize the geometry of the policy optimization problem from a functional perspective. Specifically, we prove that the objective function, as a bivariate function of the two players' policies, despite being non-convex and non-concave, satisfies a one-point strong monotonicity condition (Facchinei & Pang, 2007) at a Nash equilibrium. Thanks to such benign geometry, we prove that smooth FSP converges to a neighborhood of a Nash equilibrium at a sublinear $\widetilde{O}(1/T)$ rate, where $T$ is the number of policy iterations and $\widetilde{O}$ hides logarithmic factors. Moreover, as a byproduct of our analysis, if any of the two players deviates from the proposed algorithm, it is shown the other player that follows smooth FSP exploits such deviation by finding the best-response policy at a same sublinear rate. Such a Hannan consistency property exhibited in our algorithm is related to Hennes et al. (2020), which focus on normal-form games. Thus, our results also serve as a first step towards connecting regret between minimization in normal-form/extensive-form games and Markov games.

**Contribution.** Our contribution is two-fold. First, we propose a novel policy optimization algorithm for Markov games, which can be viewed as a generalization of FP. Second, when applied to a class of two-player zero-sum Markov games satisfying a Lipschitz regularity condition, our algorithm provably enjoys global convergence to a neighborhood of a Nash equilibrium at a sublinear rate. To the best of our knowledge, we propose the first provable FSP-type algorithm with finite time convergence guarantee for zero-sum Markov games.

**Related Work.** There is a large body of literature on the value-based methods to zero-sum Markov games (Lagoudakis & Parr, 2012; Pérolat et al., 2016; Zhang et al., 2018; Zou et al., 2019). More recently, Perolat et al. (2018) prove that actor-critic fictitious play asymptotically converges to the Nash equilibrium, while our work provides finite time convergence guarantee to a neighborhood of a Nash equilibrium. In addition, Zhang et al. (2020) study the sample comlexity of planning algorithm in the model-based MARL settting as opposed to the model-free setting with function approximation in this paper.

Closely related to smooth FSP proposed in this paper, there is a line of work in best-response algorithms (Heinrich et al., 2015; Heinrich & Silver, 2016), which have also shown great empirical performances (Dudziak, 2006; Xiao et al., 2013; Kawamura et al., 2017). However, they are only applicable to extensive-form games and not directly applicable to stochastic games. Also, our smooth FSP is related to Swenson & Poor (2019), which focus on the potential games. It does not enforce entropy-regularization and only provides asymptotic convergence guarantee to a neighborhood of the

Nash equilibrium for smooth fictitious play in multi-player two-action potential games. Moreover, our work also falls into the realm of regularizing and smoothing techniques in reinforcement learning (Dai et al., 2017; Geist et al., 2019; Shani et al., 2019; Cen et al., 2020), which focus on the single-agent setting.

## 2 BACKGROUND

In this section, we briefly introduce the general setting of reinforcement learning for two-player zero-sum Markov games.

**Zero-Sum Markov Games.** We consider the two-player zero-sum Markov game $(\mathcal{S}, \mathcal{A}^1, \mathcal{A}^2, \mathcal{P}, r, \gamma)$, where $\mathcal{S} \subset \mathbb{R}^d$ is a compact state space, $\mathcal{A}^1$ and $\mathcal{A}^2$ are finite action spaces of Player 1 and Player 2, respectively, $\mathcal{P} : \mathcal{S} \times \mathcal{S} \times \mathcal{A}^1 \times \mathcal{A}^2 \to [0, 1]$ is the Markov transition kernel, $r : \mathcal{S} \times \mathcal{A}^1 \times \mathcal{A}^2 \to [-1, 1]$ is the reward function of Player 1, which implies that the reward function of Player 2 is $-r$, and $\gamma \in (0, 1)$ is the discount factor. Let $r_1 = r$ and $r_2 = -r$ be the reward functions of Player 1 and Player 2, respectively. For notational simplicity, throughout this paper, we write Player $-i$ as Player $i$'s opponent, where $i \in \{1, 2\}$. In the rest of this paper, we omit $i \in \{1, 2\}$ where it is clear from the context. Also, we denote by $\mathbb{E}_{\pi^i, \pi^{-i}}[\cdot]$ the expectation over the trajectory induced by the policy pair $[\pi^i; \pi^{-i}]$.

Given a policy $\pi^{-i} : \mathcal{A}^{-i} \times \mathcal{S} \to [0, 1]$ of Player $-i$, the performance of a policy $\pi^i : \mathcal{A}^i \times \mathcal{S} \to [0, 1]$ of Player $i$ is evaluated by its state-value function ($V_i$-function) $V_i^{\pi^i, \pi^{-i}} : \mathcal{S} \to \mathbb{R}$, which is defined as

$$V_i^{\pi^i, \pi^{-i}}(s) = \mathbb{E}_{\pi^i, \pi^{-i}} \left[ \sum_{t=0}^{\infty} \gamma^t \cdot r_i(s_t, a_t^i, a_t^{-i}) \,\Big|\, s_0 = s \right]. \tag{2.1}$$

Correspondingly, the performance of a policy $\pi^i : \mathcal{A}^i \times \mathcal{S} \to [0, 1]$ of Player $i$ is evaluated by its action-value function ($Q_i$-function) $Q_i^{\pi^i, \pi^{-i}} : \mathcal{S} \times \mathcal{A}^i \times \mathcal{A}^{-i} \to \mathbb{R}$, which is defined by the following Bellman equation,

$$Q_i^{\pi^i, \pi^{-i}}(s, a^i, a^{-i}) = r_i(s, a^i, a^{-i}) + \gamma \cdot \mathbb{E}_{s' \sim \mathcal{P}(\cdot \,|\, s, a^i, a^{-i})} \big[ V_i^{\pi^i, \pi^{-i}}(s') \big].$$

We denote by $\nu_{\pi^i, \pi^{-i}}(s)$ and $\sigma_{\pi^i, \pi^{-i}}(s, a^i, a^{-i}) = \pi^i(a^i \,|\, s) \cdot \pi^{-i}(a^{-i} \,|\, s) \cdot \nu_{\pi^i, \pi^{-i}}(s)$ the stationary state distribution and the stationary state-action distribution associated with the policy pair $[\pi^i; \pi^{-i}]$, respectively. Correspondingly, we denote by $\mathbb{E}_{\sigma_{\pi^i, \pi^{-i}}}[\cdot]$ and $\mathbb{E}_{\nu_{\pi^i, \pi^{-i}}}[\cdot]$ the expectations $\mathbb{E}_{(s, a^i, a^{-i}) \sim \sigma_{\pi^i, \pi^{-i}}}[\cdot]$ and $\mathbb{E}_{s \sim \nu_{\pi^i, \pi^{-i}}}[\cdot]$, respectively. Throughout this paper, we denote by $\langle \cdot, \cdot \rangle$ the inner product between vectors.

Let $[\pi_*^1, \pi_*^2]$ be a Nash equilibrium of the two-player zero-sum Markov game $(\mathcal{S}, \mathcal{A}^1, \mathcal{A}^2, \mathcal{P}, r, \gamma)$, which exists (Shapley, 1953) and satisfies

$$\mathcal{J}(\pi^1, \pi_*^2) \leq \mathcal{J}(\pi_*^1, \pi_*^2) \leq \mathcal{J}(\pi_*^1, \pi^2)$$

for all policy pairs $[\pi^1; \pi^2]$. Here we define the performance function as

$$\mathcal{J}(\pi^1, \pi^2) = \mathbb{E}_{\nu^*} \big[ V_1^{\pi^1, \pi^2}(s) \big], \tag{2.2}$$

where $\nu^*$ is the stationary distribution $\sigma_{\pi_*^1, \pi_*^2}$.

**Regularized Markov Games.** Based on the definition of the two-player zero-sum Markov game $(\mathcal{S}, \mathcal{A}^1, \mathcal{A}^2, \mathcal{P}, r, \gamma)$, we define its entropy-regularized counterpart $(\mathcal{S}, \mathcal{A}^1, \mathcal{A}^2, \mathcal{P}, r, \gamma, \lambda_1, \lambda_2)$, where $\lambda_1, \lambda_2 \geq 0$ are the regularization parameters. Specifically, $(\mathcal{S}, \mathcal{A}^1, \mathcal{A}^2, \mathcal{P}, r, \gamma, \lambda_1, \lambda_2)$ is defined as the two-player general-sum Markov game with the reward function of Player $i$ replaced by its entropy-regularized counterpart $r_{(i)}^{\pi^i, \pi^{-i}} : \mathcal{S} \times \mathcal{A}^i \times \mathcal{A}^{-i} \to \mathbb{R}$, which is defined as

$$r_{(i)}^{\pi^i, \pi^{-i}}(s, a^i, a^{-i}) = r_i(s, a^i, a^{-i}) - \lambda_i \cdot \log \pi^i(a^i \,|\, s). \tag{2.3}$$

With a slight abuse of notation, we write

$$r_i^{\pi^i, \pi^{-i}}(s) = \mathbb{E}_{\pi^i, \pi^{-i}} \big[ r_i(s, a^i, a^{-i}) \big],$$
$$r_{(i)}^{\pi^i, \pi^{-i}}(s) = \mathbb{E}_{\pi^i, \pi^{-i}} \big[ r_{(i)}^{\pi^i, \pi^{-i}}(s, a^i, a^{-i}) \big] = r_i^{\pi^i, \pi^{-i}}(s) + \lambda_i \cdot H\big( \pi^i(\cdot \,|\, s) \big)$$

as the state-reward function and the entropy-regularized state-reward function, respectively. Here $H(\pi^i(\cdot \,|\, s)) = -\sum_{a^i \in \mathcal{A}^i} \pi^i(a^i \,|\, s) \cdot \log \pi^i(a^i \,|\, s)$ is the Shannon entropy. For Player $i$, the entropy-regularized state-value function ($V_{(i)}$-function) $V_{(i)}^{\pi^i, \pi^{-i}} : \mathcal{S} \to \mathbb{R}$ and the entropy-regularized action-value function ($Q_{(i)}$-function) $Q_{(i)}^{\pi^i, \pi^{-i}} : \mathcal{S} \times \mathcal{A}^i \times \mathcal{A}^{-i} \to \mathbb{R}$ are defined as

$$V_{(i)}^{\pi^i, \pi^{-i}}(s) = \mathbb{E}_{\pi^i, \pi^{-i}} \left[ \sum_{t=0}^{\infty} \gamma^t \cdot r_{(i)}^{\pi^i, \pi^{-i}}(s_t, a_t^i, a_t^{-i}) \,\middle|\, s_0 = s \right], \tag{2.4}$$

$$Q_{(i)}^{\pi^i, \pi^{-i}}(s, a^i, a^{-i}) = r_i(s, a^i, a^{-i}) + \gamma \cdot \mathbb{E}_{s' \sim \mathcal{P}(\cdot \,|\, s, a^i, a^{-i})} \left[ V_{(i)}^{\pi^i, \pi^{-i}}(s') \right], \tag{2.5}$$

respectively. By the definition of $r_{(i)}^{\pi^i, \pi^{-i}}$ in (2.3), we have that, for all policy pairs $[\pi^i; \pi^{-i}]$ and $s \in \mathcal{S}$,

$$\left| \mathbb{E}_{\pi^i, \pi^{-i}} \left[ r_{(i)}^{\pi^i, \pi^{-i}}(s, a^i, a^{-i}) \right] \right| \le 1 + \lambda_i \cdot \log |\mathcal{A}^i|,$$

which, by (2.4) and (2.5) implies that, for all policy pairs $[\pi^i; \pi^{-i}]$ and $(s, a^i, a^{-i}) \in \mathcal{S} \times \mathcal{A}^i \times \mathcal{A}^{-i}$,

$$\left| V_{(i)}^{\pi^i, \pi^{-i}}(s) \right| \le V_{(i)}^{\max} = \frac{1 + \lambda_i \cdot \log |\mathcal{A}^i|}{1 - \gamma}, \tag{2.6}$$

$$\left| Q_{(i)}^{\pi^i, \pi^{-i}}(s, a^i, a^{-i}) \right| \le Q_{(i)}^{\max} = 1 + \frac{\gamma \cdot (1 + \lambda_i \cdot \log |\mathcal{A}^i|)}{1 - \gamma}. \tag{2.7}$$

# 3 FICTITIOUS SELF-PLAY FOR ZERO-SUM MARKOV GAMES

In this section, we introduce smooth fictitious self-play (FSP) for two-player zero-sum Markov games.

## 3.1 FSP: FROM MATRIX GAMES TO MARKOV GAMES

FSP is an algorithmic framework for finding the Nash equilibria of games. It consists of two building blocks: (I) inferring the opponent's policy by playing against each other, namely fictitious play, and (II) improving the two players' policies with symmetric updating rules, namely self-play. Specifically, Player $i$ best responds to a mixed policy of Player $-i$, which is a weighted average of Player $-i$'s historical policies. Here playing a mixed policy $\overline{\pi}^{-i} = \alpha \cdot \pi^{-i} + (1 - \alpha) \cdot \pi^{-i\prime}$ means that, at the beginning of the game, the player chooses to play the policy $\pi^{-i}$ with probability $\alpha$ and play the policy $\pi^{-i\prime}$ with probability $1 - \alpha$.

FSP is originally developed for normal-form games (Von Neumann & Morgenstern, 2007; Shapley, 1953) and extensive-form games (Heinrich et al., 2015; Heinrich & Silver, 2016). In (entropy-regularized) two-player zero-sum matrix games, which are the special cases of (entropy-regularized) two-player zero-sum Markov games with $|\mathcal{S}| = 1$ and no state transition, mixing two policies $\pi^{-i}$ and $\pi^{-i\prime}$ with probabilities $\alpha$ and $1 - \alpha$, respectively, is equivalent to averaging the corresponding $Q_i$-functions, i.e.,

$$Q_i^{\pi^i, \alpha \cdot \pi^{-i} + (1-\alpha) \cdot \pi^{-i\prime}} = \alpha \cdot Q_i^{\pi^i, \pi^{-i}} + (1 - \alpha) \cdot Q_i^{\pi^i, \pi^{-i\prime}}.$$

In other words, in a two-player zero-sum matrix game, Player $i$ is equivalently best responding to a weighted average of the historical $Q_i$-functions by taking the corresponding greedy action. To generalize FSP to the two-player zero-sum Markov game $(\mathcal{S}, \mathcal{A}^1, \mathcal{A}^2, \mathcal{P}, r, \gamma)$, we propose to let Player $i$ best respond to the following weighted average of the historical marginalized $Q_{(i)}$-functions at the $t$-th iteration,

$$\overline{Q}_{t+1,(i)}(s, a^i) = (1 - \overline{\alpha}_{t,(i)}) \cdot \overline{Q}_{t,(i)}(s, a^i) + \overline{\alpha}_{t,(i)} \cdot \widetilde{Q}_{(i)}^{\pi_t^i, \pi_t^{-i}}(s, a^i), \tag{3.1}$$

where $\overline{\alpha}_{t,(i)} \in [0, 1]$ is the mixing rate. Here the marginalized $Q_{(i)}$-function $\widetilde{Q}_{(i)}^{\pi^i, \pi^{-i}}(s, a^i)$ is defined as

$$\widetilde{Q}_{(i)}^{\pi^i, \pi^{-i}}(s, a^i) = \mathbb{E}_{\pi^{-i}} \left[ Q_{(i)}^{\pi^i, \pi^{-i}}(s, a^i, a^{-i}) \right]. \tag{3.2}$$

Recursively applying the symmetric updating rule in (3.1), we obtain

$$\overline{Q}_{t+1,(i)}(s,a^i) = \sum_{\tau=0}^{t} \left\{ \left[ \overline{\alpha}_{\tau,(i)} \cdot \prod_{k=\tau+1}^{t} (1 - \overline{\alpha}_{k,(i)}) \right] \cdot \widetilde{Q}_{(i)}^{\pi_\tau^i, \pi_\tau^{-i}}(s,a^i) \right\},$$

which is the weighted average of the historical marginalized $Q_{(i)}$-functions. Here we use the convention that $\prod_{k=t+1}^{t} (1 - \overline{\alpha}_{k,(i)}) = 1$. Correspondingly, (3.1) induces the following symmetric policy updating rule,

$$\pi_{t+1}^{i,\text{best}}(a^i \mid s) = \mathbb{1}\Big( a^i = \underset{a^{i\prime} \in \mathcal{A}^i}{\operatorname{argmax}} \{ \overline{Q}_{t+1,(i)}(s, a^{i\prime}) \} \Big), \tag{3.3}$$

where the obtained policy $\pi_{t+1}^{i,\text{best}}$ best responds to $\overline{Q}_{t+1,(i)}$ defined in (3.1) by taking the corresponding greedy action.

## 3.2 Markov Games: From FSP to Smooth FSP

FSP is only known to converge asymptotically even in two-player zero-sum matrix games (Robinson, 1951). Instead, we consider smooth FSP, which uses the following smoothed best-response,

$$\pi_{t+1}^i(a^i \mid s) \propto \exp\{ \mathcal{E}_{t+1,(i)}(s, a^i) \}. \tag{3.4}$$

Here the ideal energy function $\mathcal{E}_{t+1,(i)}(s, a^i) = \kappa_{t+1,(i)} \cdot \overline{Q}_{t+1,(i)}(s, a^i)$ is proportional to the weighted average of the historical marginalized $Q_{(i)}$-functions defined in (3.1) with the normalization parameter $\kappa_{t+1,(i)} > 0$.

In the sequel, we simplify the symmetric updating rules in (3.1) and (3.4). Let the stepsizes be

$$\alpha_{t,(i)} = \kappa_{t+1,(i)} \cdot \overline{\alpha}_{t,(i)}, \quad \alpha_{t,(i)}' = \kappa_{t+1,(i)} / \kappa_{t,(i)} \cdot (1 - \overline{\alpha}_{t,(i)}). \tag{3.5}$$

Recall that $\widetilde{Q}_{(i)}^{\pi_t^i, \pi_t^{-i}}$, which is the marginalized $Q_{(i)}$-function, is defined in (3.2). Corresponding to (3.1), we have the following symmetric updating rule for the energy functions,

$$\mathcal{E}_{t+1,(i)}(s, a^i) = \alpha_{t,(i)}' \cdot \mathcal{E}_{t,(i)}(s, a^i) + \alpha_{t,(i)} \cdot \widetilde{Q}_{(i)}^{\pi_t^i, \pi_t^{-i}}(s, a^i), \tag{3.6}$$

which gives the following symmetric policy updating rule equivalent to (3.4),

$$\pi_{t+1}^i(a^i \mid s) \propto \big( \pi_t^i(a^i \mid s) \big)^{\alpha_{t,(i)}'} \cdot \exp\{ \alpha_{t,(i)} \cdot \widetilde{Q}_{(i)}^{\pi_t^i, \pi_t^{-i}}(s, a^i) \}.$$

We call $\mathcal{E}_{t+1,(i)}$ the ideal energy function, since it is directly obtained from the symmetric updating rule in (3.3), which operates in the functional space given the marginalized $Q_{(i)}$-functions.

## 3.3 Implementing Smooth FSP

In practice, it remains to approximate the ideal energy function $\mathcal{E}_{t+1,(i)}$ within a parameterized function class, which is further used to parameterize the policy $\pi_{t+1}^i$. For notational simplicity, we concatenate the parameters of the policies $\pi_{t+1}^i$ and $\pi_{t+1}^{-i}$ into a single parameter $\theta_{t+1} \in \Theta$, which gives the parameterized policy pair $[\pi_{\theta_t}^i; \pi_{\theta_t}^{-i}]$. Meanwhile, we need to estimate the marginalized $Q_{(i)}$-function $\widetilde{Q}_{(i)}^{\pi_{\theta_t}^i, \pi_{\theta_t}^{-i}}(s, a^i)$ defined in (3.2). In practice, the parameterization of the energy function and the marginalized $Q_{(i)}$-function are set to be neural networks, which means that $\Theta = \mathbb{R}^N$ with $N$ being the size of the neural network. To implement smooth FSP, given $\theta_t \in \Theta$, we find the best parameter $\theta_{t+1} \in \Theta$ that minimizes the mean squared error (MSE),

$$\mathbb{E}_{\sigma_t} \left[ \sum_{i \in \{1,2\}} \big( \mathcal{E}_{\theta_{t+1},(i)}(s, a^i) - \widehat{\mathcal{E}}_{t+1,(i)}(s, a^i) \big)^2 \right], \tag{3.7}$$

$$\text{where } \widehat{\mathcal{E}}_{t+1,(i)}(s, a^i) = \alpha_{t,(i)}' \cdot \mathcal{E}_{\theta_t,(i)}(s, a^i) + \alpha_{t,(i)} \cdot \widehat{Q}_{(i)}^{\pi_{\theta_t}^i, \pi_{\theta_t}^{-i}}(s, a^i) \tag{3.8}$$

is the estimated ideal energy function. Here $\widehat{Q}_{(i)}^{\pi_{\theta_t}^i, \pi_{\theta_t}^{-i}}(s, a^i)$ is the estimator of the marginalized $Q_{(i)}$-function $\widetilde{Q}_{(i)}^{\pi_{\theta_t}^i, \pi_{\theta_t}^{-i}}(s, a^i)$. Such an estimator is obtained based on the data generated by smooth FSP via policy evaluation (Sutton et al., 2000). For notational simplicity, in (3.7) and the rest of the paper,

we write the stationary state-action distribution $\sigma_{\pi^i_{\theta_t}, \pi^{-i}_{\theta_t}}$ and the stationary state distribution $\nu_{\pi^i_{\theta_t}, \pi^{-i}_{\theta_t}}$ associated with the policy pair $[\pi^i_{\theta_t}; \pi^{-i}_{\theta_t}]$ as $\sigma_t$ and $\nu_t$, respectively.

We define the bounded function class $\mathcal{F}_R$ with the radius $R > 0$ as $\mathcal{F}_R = \{f : \|f\|_\infty \leq R\}$. Algorithm 1 gives the implementation of smooth FSP for two-player zero-sum Markov games.

---

**Algorithm 1** Smooth FSP for Two-Player Zero-Sum Markov Games

---

1: **Require** Two-player zero-sum Markov game $(\mathcal{S}, \mathcal{A}^1, \mathcal{A}^2, \mathcal{P}, r, \gamma)$, number of iterations $T$, regularization parameters $\{\lambda_i\}_{i \in \{1,2\}}$, truncation parameters $\{Q^{\max}_{(i)}, \mathcal{E}^{\max}_{(i)}\}_{i \in \{1,2\}}$, and stepsizes $\{\alpha_{t,(i)}, \alpha'_{t,(i)}\}_{0 \leq t \leq T-1, i \in \{1,2\}}$
2: Initialize the energy function $\mathcal{E}_{\theta_0,(i)}(s, a^i) \leftarrow 0$ $(i \in \{1, 2\})$
3: **For** $t = 0, \ldots, T - 1$ and $i \in \{1, 2\}$ **do**
4:     Set the policy $\pi^i_{\theta_t}(\cdot \mid s) \propto \exp\{\mathcal{E}_{\theta_t,(i)}(s, \cdot)\}$
5:     Generate the marginalized $Q_{(i)}$-function estimator $\widehat{Q}^{\pi^i_{\theta_t}, \pi^{-i}_{\theta_t}}_{(i)}(s, a^i) \in \mathcal{F}_{Q^{\max}_{(i)}}$ using the data generated by fictitious play with the policy pair $[\pi^i_{\theta_t}; \pi^{-i}_{\theta_t}]$
6:     Update the estimated ideal energy function

$$\widehat{\mathcal{E}}_{t+1,(i)}(s, a^i) \leftarrow \alpha'_{t,(i)} \cdot \mathcal{E}_{\theta_t,(i)}(s, a^i) + \alpha_{t,(i)} \cdot \widehat{Q}^{\pi^i_{\theta_t}, \pi^{-i}_{\theta_t}}_{(i)}(s, a^i)$$

7:     Minimize (3.7) to obtain the energy function $\mathcal{E}_{\theta_{t+1},(i)}(s, a^i) \in \mathcal{F}_{\mathcal{E}^{\max}_{(i)}}$
8: **End**
9: **Output:** $\{\pi^i_{\theta_t}\}_{0 \leq t \leq T-1, i \in \{1,2\}}$

---

# 4 MAIN RESULTS

In this section, we establish the convergence of smooth FSP for two-player zero-sum Markov games by casting it as regularized proximal policy optimization (PPO).

## 4.1 SMOOTH FSP AS REGULARIZED PPO

In the sequel, we connect the energy function update in (3.8) with regularized PPO. Corresponding to the estimated ideal energy function updates $\widehat{\mathcal{E}}_{t+1,(i)}$ in (3.8), we define the estimated ideal policy update as

$$\widehat{\pi}^i_{t+1}(\cdot \mid s) \propto \exp\{\widehat{\mathcal{E}}_{t+1,(i)}(s, \cdot)\}. \tag{4.1}$$

We have the following proposition states the equivalence between smooth FSP and regularized PPO.

**Proposition 4.1.** For all $0 \leq t \leq T - 1$, let the stepsizes $\alpha_{t,(i)}$ and $\alpha'_{t,(i)}$ of Algorithm 1 satisfy

$$\lambda_i = (1 - \alpha'_{t,(i)})/\alpha_{t,(i)} > 0.$$

At the $t$-th iteration of Algorithm 1, the policy update in (4.1) is equivalent to solving the regularized PPO subproblem,

$$\widehat{\pi}^i_{t+1} = \underset{\pi^i}{\arg\max}\Bigg\{ \mathbb{E}_{\nu_t}\Big[\alpha_{t,(i)} \cdot \big\langle \widehat{Q}^{\pi^i_{\theta_t}, \pi^{-i}_{\theta_t}}_{(i)}(s, \cdot) - \lambda_i \cdot \log \pi^i_{\theta_t}(\cdot \mid s), \pi^i(\cdot \mid s) - \pi^i_{\theta_t}(\cdot \mid s)\big\rangle \tag{4.2}$$

$$- \mathrm{KL}\big(\pi^i(\cdot \mid s) \,\big\|\, \pi^i_{\theta_t}(\cdot \mid s)\big)\Big]\Bigg\}.$$

Here $\widehat{Q}^{\pi^i_{\theta_t}, \pi^{-i}_{\theta_t}}_{(i)}(s, a^i)$ is the estimator of the marginalized $Q_{(i)}$-function $\widetilde{Q}^{\pi^i_{\theta_t}, \pi^{-i}_{\theta_t}}_{(i)}(s, a^i)$.

*Proof.* See Appendix A for a detailed proof. $\qquad\square$

Proposition 4.1 implies that smooth FSP proximally improves the policy $\pi^i$ based on the regularized performance function,

$$\mathcal{J}_{(i)}(\pi^i, \pi^{-i}) = \mathbb{E}_{\nu^*}\big[V_{(i)}^{\pi^i, \pi^{-i}}(s)\big]. \tag{4.3}$$

Proposition C.1 implies that, the smaller the regularization parameter $\lambda_i$ is, the closer the regularized performance function $\mathcal{J}_{(i)}$ is to the performance function $\mathcal{J}$. In the rest of the paper, we show that, with a proper choice of $\lambda_i$, smooth FSP converges to a neighborhood of a Nash equilibrium $[\pi_*^1; \pi_*^2]$ at a sublinear rate of $\widetilde{O}(1/T)$.

## 4.2 Convergence to Nash Equilibrium

Let $\mathbb{P}(s_t = s \mid \pi^i, \pi^{-i}, s_0 \sim \nu)$ be the probability that the trajectory, which is generated by the policy pair $[\pi^i; \pi^{-i}]$ with the initial state distribution $s_0 \sim \nu$, reaches the state $s$ at the timestep $t$. Correspondingly, let

$$\rho_\nu^{\pi^i, \pi^{-i}}(s) = (1 - \gamma) \cdot \sum_{t=0}^\infty \gamma^t \cdot \mathbb{P}(s_t = s \mid \pi^i, \pi^{-i}, s_0 \sim \nu) \tag{4.4}$$

be the visitation measure of $[\pi^i; \pi^{-i}]$ with the initial state distribution $s_0 \sim \nu$. Also, for notational simplicity, we define

$$\rho_{\nu, \pi^{i\prime}, \pi^{-i\prime}}^{\pi^i, \pi^{-i}}(s) = (1 - \gamma) \cdot \sum_{t=0}^\infty \gamma^t \cdot \mathbb{P}\big(s_{t+1} = s \mid \pi^i, \pi^{-i}, (s_0, a_0^i, a_0^{-i}) \sim \nu\pi^{i\prime}\pi^{-i\prime}\big) \tag{4.5}$$

as the visitation measure of the policy pair $[\pi^i; \pi^{-i}]$ with the initial state-action distribution $\nu\pi^{i\prime}\pi^{-i\prime}$. We lay out the following assumption on the concentrability coefficient. With a slight abuse of notation, we write $\nu$ and $\pi^{i\prime}$ in the subscripts as $s$ and $a^i$, respectively, when they are point masses.

**Assumption 4.2** (Concentrability Coefficient). We assume that for the two-player zero-sum Markov game $(\mathcal{S}, \mathcal{A}^1, \mathcal{A}^2, \mathcal{P}, r, \gamma)$, there exists $\zeta > 0$ such that

$$\mathbb{E}_{\nu^*}\Big[\big|\mathrm{d}\rho_{s, a^i, \pi_*^{-i}}^{\pi^i, \pi_*^{-i}}/\mathrm{d}\nu^*\big|^2\Big]^{1/2} \le \zeta$$

for all $s \in \mathcal{S}$, $a^i \in \mathcal{A}^i$, and $\pi^i = \pi_{\theta_t}^i$ generated by the policy update in Line 4 of Algorithm 1. Here $\mathrm{d}\rho_{s, a^i, \pi_*^{-i}}^{\pi^i, \pi_*^{-i}}/\mathrm{d}\nu^*$ is the Radon-Nikodym derivative, where $\rho_{s, a^i, \pi_*^{-i}}^{\pi^i, \pi_*^{-i}}$ is defined in (4.5).

The notion of concentrability coefficient in Assumption 4.2 is commonly used in the literature (Munos & Szepesvári, 2008; Antos et al., 2008; Farahmand et al., 2010; Tosatto et al., 2017; Yang et al., 2019).

For all policy pairs $[\pi^i; \pi^{-i}]$, we define the Markov state transition kernel as

$$\mathcal{P}^{\pi^i, \pi^{-i}}(\cdot \mid s) = \mathbb{E}_{\pi^i, \pi^{-i}}\big[\mathcal{P}(\cdot \mid s, a^i, a^{-i})\big]. \tag{4.6}$$

With a slight abuse of notation, we write $\mathcal{P}^{\pi^i, \pi^{-i}}$ as the Markov state transition operator induced by the Markov state transition kernel defined in (4.6), such that

$$[\mathcal{P}^{\pi^i, \pi^{-i}} \circ h](s) = \int_{s' \in \mathcal{S}} h(s')\mathcal{P}^{\pi^i, \pi^{-i}}(\mathrm{d}s' \mid s), \tag{4.7}$$

where $h : \mathcal{S} \to \mathbb{R}$ is an $L_1$-integrable function and the Lebesgue measure over $\mathcal{S} \subset \mathbb{R}^d$ is used. Correspondingly, we define the operator norm of an operator $\mathcal{O}$ as

$$\|\mathcal{O}\|_{\mathrm{op}} = \sup_h \|\mathcal{O} \circ h\|_{L_1(\mathcal{S})}\big/\|h\|_{L_1(\mathcal{S})} = \sup_{\|h\|_{L_1(\mathcal{S})} \le 1} \|\mathcal{O} \circ h\|_{L_1(\mathcal{S})},$$

where $\|\cdot\|_{L_1(\mathcal{S})}$ is the $L_1$-norm over the state space $\mathcal{S}$. The following assumption characterizes the Lipschitz continuity of $\mathcal{P}^{\pi^i, \pi^{-i}}$ and $r^{\pi^i, \pi^{-i}}$ with respect to $\pi^{-i}$.

**Assumption 4.3** (Lipschitz Game). We assume that for the two-player zero-sum Markov game $(\mathcal{S}, \mathcal{A}^1, \mathcal{A}^2, \mathcal{P}, r, \gamma)$, there exists $\iota_i > 0$ such that for all $s \in \mathcal{S}$ and $[\pi^i; \pi^{-i}]$,

$$\|\mathcal{P}^{\pi^i, \pi_*^{-i}} - \mathcal{P}^{\pi^i, \pi^{-i}}\|_{\mathrm{op}} \le \iota_i \cdot \mathbb{E}_{\nu^*}\big[\mathrm{KL}\big(\pi_*^{-i}(\cdot \mid s) \,\big\|\, \pi^{-i}(\cdot \mid s)\big)\big]^{1/2}, \tag{4.8}$$

$$\big|r^{\pi^i, \pi_*^{-i}}(s) - r^{\pi^i, \pi^{-i}}(s)\big| \le \iota_i \cdot \mathrm{KL}\big(\pi_*^{-i}(\cdot \mid s) \,\big\|\, \pi^{-i}(\cdot \mid s)\big)^{1/2}. \tag{4.9}$$

The Lipschitz coefficient $\iota_i$ in (4.8) of Assumption 4.3 quantifies to the influence of Player $-i$ on the nonstationary environment that Payer $i$ faces. Such a notion of influence is proposed by Radanovic et al. (2019) in the tabular setting. In particular, the expected KL-divergence between the policies is used in place of the distance $\max_{s\in\mathcal{S}} \|\pi^{-i}(\cdot\,|\,s) - \pi^{-i\prime}(\cdot\,|\,s)\|_1$ in Radanovic et al. (2019). Such an assumption is also related to the linear-quadratic game (LQG) literature (see, e.g., Zhang et al. (2019)), where the Lipschitz continuity is established based on the special structure in the LQG model. In Lemma C.2, we show that such a Lipschitz coefficient $\iota_i$ quantifies the Lipschitz continuity of the marginalized $Q_{(i)}$-function of the entropy-regularized two-player Markov game $(\mathcal{S}, \mathcal{A}^1, \mathcal{A}^2, \mathcal{P}, r, \gamma, \lambda_1, \lambda_2)$.

Recall that $\widehat{\pi}_{t+1}^i \propto \exp\{\widehat{\mathcal{E}}_{t+1,(i)}\}$ is defined in (4.1), where $\widehat{\mathcal{E}}_{t+1,(i)}$ is defined in (3.8). Also, recall that $\pi_{\theta_{t+1}}^i \propto \exp\{\mathcal{E}_{\theta_{t+1},(i)}\}$ is defined in Line 4 of Algorithm 1, where $\mathcal{E}_{\theta_{t+1},(i)}$ is obtained by minimizing (3.7) in Line 7 of Algorithm 1. Meanwhile, we define the ideal policy update as
$$\overline{\pi}_{t+1}^i(\cdot\,|\,s) \propto \exp\{\overline{\mathcal{E}}_{t+1,(i)}(s,\cdot)\},$$
where $\overline{\mathcal{E}}_{t+1,(i)}(s,a^i) = \alpha'_{t,(i)} \cdot \mathcal{E}_{\theta_t,(i)}(s,a^i) + \alpha_{t,(i)} \cdot \widetilde{Q}_{(i)}^{\pi_{\theta_t}^i, \pi_{\theta_t}^{-i}}(s,a^i)$ (4.10)
is the corresponding ideal energy function update.

We lay out the following assumption on the errors that arise from the estimation of the marginalized $Q_{(i)}$-function $\widetilde{Q}_{(i)}^{\pi_{\theta_t}^i, \pi_{\theta_t}^{-i}}$ and the minimization of the MSE in (3.7).

**Assumption 4.4** (Estimation Error). We assume that there exist $\epsilon_t, \epsilon_t' > 0$ such that for all $0 \le t \le T-1$,
$$\mathbb{E}_{\nu^*}\Big[\big\|\mathcal{E}_{\theta_{t+1},(i)}(s,\cdot) - \widehat{\mathcal{E}}_{t+1,(i)}(s,\cdot)\big\|_\infty^2\Big] \le \epsilon_t, \tag{4.11}$$
$$\Big|\mathbb{E}_{\nu^*}\Big[\big\langle \mathcal{E}_{\theta_{t+1},(i)}(s,\cdot) - \overline{\mathcal{E}}_{t+1,(i)}(s,\cdot), \pi_*^i(\cdot\,|\,s) - \pi_{\theta_t}^i(\cdot\,|\,s)\big\rangle\Big]\Big| \le \epsilon_t'. \tag{4.12}$$

Assumption 4.4 characterizes the estimation error through the policy updates in Line 7 of Algorithm 1. In particular, (4.11) upper bounds the errors arising from the minimization of the MSE in (3.7), which is zero as long as the representation power of the parameterized class of the energy functions is sufficiently strong. Meanwhile, by (3.7) and (4.10), the gap between $\mathcal{E}_{\theta_{t+1},(i)}$ and $\overline{\mathcal{E}}_{t+1,(i)}$ involves (I) the gap between $\widehat{\mathcal{E}}_{t+1,(i)}$ and $\overline{\mathcal{E}}_{t+1,(i)}$, which arises from the gap between $\widehat{Q}_{(i)}^{\pi_{\theta_t}^i, \pi_{\theta_t}^{-i}}$ and $\widetilde{Q}_{(i)}^{\pi_{\theta_t}^i, \pi_{\theta_t}^{-i}}$, and (II) the gap between $\mathcal{E}_{\theta_{t+1},(i)}$ and $\widehat{\mathcal{E}}_{t+1,(i)}$, which arises the minimization of the MSE in (3.7). Hence, $\epsilon_t'$ in (4.12) is zero as long as the estimator $\widehat{Q}_{(i)}^{\pi_{\theta_t}^i, \pi_{\theta_t}^{-i}}$ of $\widetilde{Q}_{(i)}^{\pi_{\theta_t}^i, \pi_{\theta_t}^{-i}}$ is accurate and $\epsilon_t$ is zero.

We summarize $\epsilon_t$ and $\epsilon_t'$ into the following total error $\sigma$,
$$\sigma = \sum_{t=0}^{T-1} (t+1) \cdot (\epsilon_t + \epsilon_t'). \tag{4.13}$$
As discussed in Lemmas 4.7 and 4.8 of Liu et al. (2019), under Assumption 4.2, when we use sufficiently deep and wide neural networks equipped with the rectified linear unit (ReLU) activation function to parameterize the marginalized $Q_{(i)}$-functions and the energy functions, Assumption 4.4 can be satisfied with $\sigma = \widetilde{O}(1)$. See Appendix B for a detailed discussion.

We are now ready to present the following theorem on the convergence of the policy sequence $\{[\pi_{\theta_t}^1; \pi_{\theta_t}^2]\}_{0\le t\le T-1}$ to a neighborhood of a Nash equilibrium $[\pi_*^1; \pi_*^2]$. Recall that $Q_{(i)}^{\max}$ and $V_{(i)}^{\max}$ are defined in (2.6) and (2.7), respectively. Also, recall that $\zeta$ is the concentrability coefficient in Assumption 4.2, $\iota_i$ is the Lipschitz coefficient in Assumption 4.3, and $\sigma$ is defined in (4.13).

**Theorem 4.5** (Convergence of Smooth FSP to Nash Equilibrium). Suppose that Assumptions 4.2-4.4 hold. We set the regularization parameter $\lambda_i \ge 2M_i$, where
$$M_i = \Big[2 + \sum_{i\in\{1,2\}} (V_{(i)}^{\max} + Q_{(i)}^{\max} \cdot \zeta)/(1-\gamma)\Big] \cdot \iota_i. \tag{4.14}$$
In Algorithm 1, we set $\mathcal{E}_{(i)}^{\max} = Q_{(i)}^{\max}/(\lambda_i - M_i)$ and
$$\alpha_{t,(i)} = \frac{1}{(t+1)\cdot\min_{i\in\{1,2\}}\{\lambda_i - M_i\}}, \quad \alpha'_{t,(i)} = 1 - \frac{\lambda_i}{(t+1)\cdot\min_{i\in\{1,2\}}\{\lambda_i - M_i\}}. \tag{4.15}$$

For the policy sequence $\{[\pi^1_{\theta_t}; \pi^2_{\theta_t}]\}_{0 \leq t \leq T-1}$ generated by the policy update in Line 7 of Algorithm 1, we have

$$\frac{1}{T} \cdot \sum_{t=0}^{T-1} \left[ \mathcal{J}(\pi^1_*, \pi^2_{\theta_t}) - \mathcal{J}(\pi^1_{\theta_t}, \pi^2_*) \right] \leq \frac{\sum_{i \in \{1,2\}} \left[ 2 + 2\lambda_i^2/(\lambda_i - M_i)^2 \right] \cdot (Q^{\max}_{(i)})^2}{(1 - \gamma) \cdot \min_{i \in \{1,2\}} \{\lambda_i - M_i\}} \cdot \frac{\log T}{T} \quad (4.16)$$

$$+ \frac{2\sigma \cdot \min_{i \in \{1,2\}} \{\lambda_i - M_i\}}{(1 - \gamma) \cdot T} + \sum_{i \in \{1,2\}} \lambda_i \cdot \log |\mathcal{A}^i|.$$

*Proof.* See Appendix C for a detailed proof. The key to our proof is the convergence of infinite-dimensional mirror descent with the primal and dual errors. In particular, the errors are characterized in Appendix B. □

Recall that the Lipschitz coefficient $\iota_i$ is defined in Assumption 4.3. In Lemma C.2, we interpret $\iota_i$ as the Lipschitz coefficient of the marginalized $Q_{(i)}$-function. Meanwhile, recall that Theorem 4.5 requires $\lambda_i \geq 2M_i$, where $M_i$ scales linearly with $\iota_i$. Hence, the smaller the Lipschitz coefficient $\iota_i$ is, the smaller the regularization parameter $\lambda_i$ can be, which in turn leads to a smaller regularization bias characterized in Proposition C.1. Thus, the policy sequence $\{[\pi^1_{\theta_t}; \pi^2_{\theta_t}]\}_{0 \leq t \leq T-1}$ generated by Algorithm 1 converges to a smaller neighborhood of a Nash equilibrium $[\pi^1_*; \pi^2_*]$.

We give the following two sufficient conditions for the Lipschitz coefficients. (I) The two players have similar influence to the game, i.e., $\iota_1/\iota_2 = O(1)$: a sufficient requirement on both of the Lipschitz coefficients is

$$\iota_i \leq (1 - \gamma)^2 / \left[ 8(1 + \gamma) \cdot \log |A^i| \right], \quad i \in \{1, 2\}.$$

(II) One of the two players (without loss of generality, we assume it is Player 2) has dominant influence to the game compared to the other: let $\iota_1/\iota_2 = z > 0$, in which case we set $M_i$ in (4.14) as

$$M_i = \sqrt{2z} \cdot \left[ 2 + \sum_{i \in \{1,2\}} (V^{\max}_{(i)} + Q^{\max}_{(i)} \cdot \zeta)/(1 - \gamma) \right] \cdot \iota_2, \quad \{1, 2\}.$$

Then one sufficient requirement on the ratio $z$ is

$$z \leq (1 - \gamma)^4 / \left[ 16(1 + \gamma)\iota_2 \cdot \log(|\mathcal{A}^1| \cdot |\mathcal{A}^2|) \right]^2.$$

As $z$ moves towards zero, the convergence guarantee approaches those for single-controller case. Please see Appendix I for a more detailed illustration on case(II).

We remark in the following that, with stronger assumptions, we can strengthen Theorem 4.5 to satisfy Hannan consistency.

**Remark 4.6** (Hannan Consistency). When Assumptions 4.2-4.4 hold for any policy $[\pi^{i\prime}; \pi^{-i\prime}]$ instead of only a Nash equilibrium $[\pi^i_*; \pi^{-i}_*]$, we can prove that, when one of the player does not update the policy as described in Algorithm 1, the opposing player can exploit the strategies it plays. Specifically, for example, when Player 2 plays the policy sequence $\{\widetilde{\pi}^2_t\}_{0 \leq t \leq T-1}$ while Player 1 updates its policy according to Algorithm 1, we have

$$\sup_{\pi^1} \left\{ \frac{1}{T} \cdot \sum_{t=0}^{T-1} \left[ \mathcal{J}(\pi^1, \widetilde{\pi}^2_t) - \mathcal{J}(\pi^1_{\theta_t}, \widetilde{\pi}^2_t) \right] \right\} \quad (4.17)$$

$$\leq \frac{\sigma \cdot (\lambda_1 - M_1)}{(1 - \gamma) \cdot T} + \frac{\left[ 2 + 2\lambda_1^2/(\lambda_1 - M_1)^2 \right] \cdot (Q^{\max}_{(1)})^2}{(1 - \gamma) \cdot (\lambda_1 - M_1)} \cdot \frac{\log T}{T} + \lambda_1 \cdot \log |\mathcal{A}^1|,$$

which implies that the policy sequence $\{\pi^1_{\theta_t}\}_{0 \leq t \leq T-1}$ converges to the best policy in hindsight with respect to $\{\widetilde{\pi}^2_t\}_{0 \leq t \leq T-1}$. As a consequence, we can also replace the left-hand side of (4.16) by the following duality gap,

$$\sup_{\pi^1} \left\{ \frac{1}{T} \cdot \sum_{t=0}^{T-1} \mathcal{J}(\pi^1, \pi^2_{\theta_t}) \right\} - \inf_{\pi^2} \left\{ \frac{1}{T} \cdot \sum_{t=0}^{T-1} \mathcal{J}(\pi^1_{\theta_t}, \pi^2) \right\}. \quad (4.18)$$

See Appendix J for a more detailed illustration on Remark 4.6.

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
