# OpenReview forum: "Policy Optimization in Zero-Sum Markov Games: Fictitious Self-Play Provably Attains Nash Equilibria"
_ICLR.cc/2021/Conference — Reject_

### Official Review · AnonReviewer3 · 2020-10-28
**Well written and interesting paper**

**Rating:** 6
**Confidence:** 4

**Review:**

This paper studies the two-player zero-sum Markov game using fictitious self-play (FSP) strategies. The authors proposed a novel entropy regularized policy optimization method for both agents. They proved the sequence of joint policies converges to a neighborhood of a Nash equilibrium at a sublinear rate. The paper is well written though the notations are a little bit complicated for readers to understand. The results seem to be rigorous.

One drawback is that the proposed algorithm is not evaluated using any empirical studies. Since the algorithm is new to the literature, it would be expected to see how it performs compared with other baseline methods in experiments.

Have you considered the stochastic variance reduced policy gradient methods? There has been an active line of work (see [1-5] for some examples) that shows the variance reduction techniques can improve the convergence rate of policy optimization methods in the single-agent setting. It would be interesting to know whether the convergence of the smooth FSP can be also improved using the same techniques.

[1] Papini M, Binaghi D, Canonaco G, Pirotta M, Restelli M. Stochastic Variance-Reduced Policy Gradient. InInternational Conference on Machine Learning 2018.
[2] Xu P, Gao F, Gu Q. An improved convergence analysis of stochastic variance-reduced policy gradient. In Uncertainty in Artificial Intelligence 2019.
[3] Shen Z, Ribeiro A, Hassani H, Qian H, Mi C. Hessian aided policy gradient. In International Conference on Machine Learning 2019.
[4] Xu, P., Gao, F. and Gu, Q. Sample Efficient Policy Gradient Methods with Recursive Variance Reduction. In International Conference on Learning Representations 2020.
[5] Huang F, Gao S, Pei J, Huang H. Momentum-Based Policy Gradient Methods. In International Conference on Machine Learning 2020.

Equation (2.3) is not entropy regularized. Instead, the state-reward function is entropy regularized.

How is the mean squared error in (3.7) solved? In the proposed algorithm, it is assumed that this can be exactly solved. However, a practical approximation of this solution will cause additional estimation error. As is required in Equation (4.13), it seems that the authors assume the estimation error to be roughly in the order of 1/t^2. I am not sure whether this strong convergence can be established using sampled data for (3.7).

The convergence result in Theorem 4.5 is upper bounded by a very large term \lambda_i \log|A^i|, where \lambda_i is larger than the Lipschitz constant, and |A^i| is the size of the action space. If I understand it correctly, both quantities are nonvanishing and thus the result in Theorem 4.5 is not convergent. I did not see any discussion in the paper to address this issue or discuss how the neighborhood can be shrunk to a smaller region.

---

> ### Author Response · Authors · 2020-11-22
> **Review Response**
>
>
> Thank you for your valuable feedback.
>
> 1. Stochastic variance reduced policy gradient methods: That would be a very interesting direction for our future work. We would expect that, with additional variance reduction techniques incorporated to our algorithm, the part with the term $\sigma$ in our convergence guarantee can be improved since MSE is reduced. We added this direction in our introduction as related works and potential improvements over our algorithm.
>
> 2. Eq(2.3), regularized reward function: We call such a state-action reward function $r^{\pi^i, \pi^{-i}}(s, a^i, a^{-i})$ as "regularized" since the fact that
> $$
> \sum_{i \in A^i} \pi^i(a^i|s)\log \pi^i(a^i|s) = - H(\pi^i(\cdot|s))
> $$
> makes the corresponding state reward function $r^{\pi^i, \pi^{-i}}(s, a^i)$ entropy regularized, which is standard in regularized MDP literature (please see, e.g., [4,5]).
>
> 3. Estimation errors: The discussion on the estimation of the marginalized action-value function and the policy update is in Appendix B. As shown in some recent advances in theories of RL with function approximations [1,2,3], the desired accuracy can be achieved if both the policies and the marginalized action-value functions are parameterized by overparameterized neural-networks. Here the $1/t^2$ is the order of required estimation error at iteration t, not the convergence rate of the estimation algorithm itself. As a side note, the approximation of action-value function via neural networks works the same no matter the MDP is entropy regularized or not since in both cases there are valid (regularized) Bellman equations.
>
> 4. Term $\lambda_i |A^i|$: We've added more elaboration on this term in our revision. In short, if the Lipschitz coefficients $\iota_i$ of both players are sufficiently small, or if the ratio between their Lipschitz constants is sufficiently small (meaning one player has much more influence to the game than the other), we can set small $\lambda_i$ and obtain a small neighborhood of a Nash equilibrium.
>
> 5. Experiments: We would very like to test how smooth FSP works on zero-sum Markov games and will certainly put it in our future works on this topic.
>
>
> [1] Liu, Boyi, et al. "Neural trust region/proximal policy optimization attains globally optimal policy." Advances in Neural Information Processing Systems. 2019.
>
> [2] Xu, Pan, and Quanquan Gu. "A finite-time analysis of Q-learning with neural network function approximation." arXiv preprint arXiv:1912.04511 (2019).
>
> [3] Cai, Qi, et al. "Neural temporal-difference learning converges to global optima." Advances in Neural Information Processing Systems. 2019.
>
> [4] Farahmand, Amir-massoud, et al. "Regularized policy iteration with nonparametric function spaces." The Journal of Machine Learning Research 17.1 (2016): 4809-4874.
>
> [5] Geist, Matthieu, Bruno Scherrer, and Olivier Pietquin. "A theory of regularized markov decision processes." arXiv preprint arXiv:1901.11275 (2019).

---

### Official Review · AnonReviewer1 · 2020-10-29
**Review: Policy Optimization in Zero-Sum Markov Games: Fictitious Self-Play Provably Attains Nash Equilibria**

**Rating:** 5
**Confidence:** 3

**Review:**

The authors consider self-play in zero-sum discounted two-player Markov games with compact state space and finite actions. They present a smooth fictitious self-play algorithm where each player adopts an entropy-regularized policy optimization method with the average of the past generated Q-values. Under appropriate assumptions, among which a Lipschitz regularity of the Markov game, the authors prove that this algorithm approximates the Nash equilibrium at a rate O(1/T) where T is the number of iteration.

-Contributions

-algorithmic: Smooth FSP algorithm a smooth version of fictitious self-play.
-theoretical: Convergence rate of order O(1/T) of Smooth FSP under appropriate conditions.

-Score justification/Main comments

The paper is well written. The proofs seem correct but I did not check everything in detail (see specific comments). My main concern is that the different assumptions made are a bit ad hoc (sometimes the assumption relates directly on the sequence of policies generated by the algorithm). And thus it is hard to assess if the provided bound is relevant or a trivial consequence of the assumptions (see specific comments below). As a sanity check and for a simpler proof it could be interesting to first present and analyze Smooth FSP without the estimation and approximation part first.
In fictitious play, each player plays the best response against the average of the past policies played by the adversary. Here it is not really the case since the policy used by one player is a  (smooth) best-response against a weighted average of the past Q-value, which depends also on the policy played by that player. Thus the link with Fictitious play is not completely clear.
I’m also curious about the reduction of the presented algorithm to matrix game. Do we recover a known algorithm, and what can we say about the convergence rate of the algorithm?



-Detailed comments
P2: “remains less less understood” and what do you mean by “classical optimization”?

P4, Section 3.1: the mixed policy as you defined it is not a policy (you cannot express as a certain function of s) thus talking about its Q-value does not make sense.

P5: Is the normalization parameter \kappa_{t+1,(i)} a learning rate of the normalization constant such that the probability sum to one? In the second case, it should be additive.

P6: I do not understand the last sentence before Section 3.3 what do you mean by obtained from (3.3) and “which operates in the functional space given the marginalized”?

P5, Section 3.3: \Theta is not defined, how do you parameterized \cE_{\theta} exactly? What do you mean by “the estimator of the marginalized[...]”, how do you construct it?

P13, Appendix A: maybe you should say that you consider the Lagrangian on the constrained optimization problem and if it the case also add the constraint on the fact that the \pi(a|s)\geq 0.

P6: In (4.3), you mean when \nu_t is close enough to \nu^*?

P6, Assumption 4.2: could you provide a non-trivial example where this assumption is correct? Furthermore, the assumption is made on the algorithm that you propose rather than on the model is very suspicious. In particular, since the sequence \pi_{\theta^t} is a random sequence (because based on estimated quantities) in which sense the inequality holds? Almost surely?

P7, (4.7): h is an integrable function with respect to which measure? And similarly, the L1-norm is defined with which measure?

P7, Assumption 4.3: Could you provide a non-trivial example where this assumption is correct? And this assumption is not weaker than the one proposed by Radanovi et al because the quantities E_{\nu^*}[ KL(…)] and max_{s} || ...||_1 are not comparable.

P7, Assumption 4.4: Again since you are manipulating random quantities you should precise about what you mean by this inequality. Furthermore, there is in fact no assumption here but just introducing the notations \epsilon_t and \epsilon_t’ (if we allow them to be in \bar{R}). The assumption would rather be that \sigma = O(1). And I’m not totally convinced it is a reasonable assumption. For example in (4.1) if \nu^t is singular with respect to \nu^* then the MSE computed with \nu_t will provide no information for the state where \nu^* is supported.

P8, Theorem 4.5: In fact M_i depends on \lambda_i trough V_{(i)}^{max}…, thus it is not clear at all if you can set \lambda_i \geq 2M_i. In fact, you are adding an additional constraint on the different parameters. Could you make this explicit in the statement of the theorem?

P16: In (C.10) in seems that you used t+1-(\lambda_i+M_i)/(max_{i} \lambda_i+M_i) \leq t which is wrong.

P16: In (C.11) the left-hand side should be divided by T.

P23: In (G.2) it should be \log(\bar{\pi}^i_{t+1}) instead of \log(\pi^i_{t+1})?

P23, end of the proof of Lemma C.4: it is \tilde{Q}.

---

> ### Author Response · Authors · 2020-11-22
> **Review Response**
>
> Thank you for your valuable feedback.
>
>
> 1. Link to FSP: Since FSP was previously only designed for potential games and extensive-form games, the smooth FSP introduced in this paper is designed for Markov games and thus does not directly compare to them. The idea of smooth FSP stems from the observation that mixing policies is equivalent to averaging their action-value functions.
>
> 2. Recovery of known algorithms: If we only have a single state (i.e., the game degenerates to a matrix/potential game), smooth FSP recovers smooth FP in matrix/potential games (please see, e.g., [1,2]).
>
> 3. P2, classical optimization: "Classical optimization" refers to the works in the optimization community. Due to the hardness of solving non-convex non-concave minimax problem in general, progress remains very limited in the optimization community even without function approximation.
>
> 4. P4, Section 3.1: By the definition of mixed policies, the action-value function for a player equals to the weighted average of the marginalized action-value functions corresponding to the components in the mixed policy. As stated in the second paragraph of Section 3.1, the observation
> $$
> Q_i^{\pi^i, \alpha\cdot \pi^{-i} + (1 - \alpha) \cdot {\pi^{-i\prime}}} = \alpha \cdot Q_i^{\pi^{i} , \pi^{-i}} + (1 - \alpha)\cdot Q_i^{{\pi^{i}}, {\pi^{-i\prime}}}.
> $$
> serves as our motivation behind the design of smooth FSP for Markov games (i.e., averaging $Q_{(i)}$-functions) and we do not talk about any mixed policy's Q-value thereafter.
>
> 5. P6, ideal energy function: Given the marginalized action-value function $\tilde{Q}$, the ideal energy function is obtained via directly adding $\tilde{Q}$ to the previous energy function $\mathcal{E}_{\theta_t}$, which is an operation on the functional space containing all possible energy functions. In comparison to such operation in functional space, when implementing smooth FSP, we refit the energy function within the parameterized function class $ \mathcal{E}_\theta, \theta \in \Theta $, which is an operation in the parameter space.
>
> 6. P5, Section 3.3: We've added more specific description on the parameterization in the revised version. As mentioned in Appendix B, we would normally consider approximation via sufficiently deep and wide neural networks and $\Theta$ is the corresponding parameter space with dimension matching the size of the neural network.
>
> 7. P5, normalization parameter: The normalization parameter $\kappa$ act as a temperature parameter. Such a parameter is unspecified in this paper since it solely serves the purpose of building connection between smooth FSP and regularized PPO.
>
> 8. P6, Eq(4.3): Here we do not mean $\nu^*$ is close to $\nu_t$. We define $\mathcal{J}_{(i)}$ as an imaginary objective that we use to track the progress of the algorithm in our analysis. Such an imaginary objective is not intended for use in practical algorithm.
>
> 9. P6, Assumption 4.2: Such a concentrability assumption is common in RL analysis literature (e.g., the related references included in our paper and [3,4]), where in many of them, such a kind of assumption is made over all policies. Non-trivial examples in single-agent setting can be found in [8], which assumes
> $$
> C_\mu = \sup_{s, a}\|d P(\cdot|s, a)/ d\mu| < \infty,
> $$
> where $\mu$ is some fixed distribution over $\mathcal{S}$. The inequality in Assumption 4.2 holds for a population quantity given the policy sequence generated by the algorithm.
>
> 10. P7, (4.7): L1-norm is defined with respect to Lebesgue measure over the state space, which is a compact subspace of $R^d$. The function $h$ is L1-integrable over the state space.
>
> 11. P7 Assumption 4.3: Thanks for pointing this out. We made an imprecise statement and have revised the explanation under this assumption. An example for it to hold would be: Suppose that the two players both have their own hidden states, which are only meaningful for themselves. This would give the state decomposition $s_t = (s^1_t, s^2_t)$, where $s^i_t$ is the hidden state of Player $i$ at timestep $t$. In this case, we can set the transition of hidden states to be an average between a transition within the hidden state space and an interactive transition that involves both players' hidden states.
> $$
> \mathcal{P}(s_{t+1}|s_t, a^1_t, a^2_t) = (1 - w) \cdot \mathcal{P}^1(s^1_{t+1}|s^1_t, a^1_t)\cdot\mathcal{P}^2(s^2_{t+1}| s^2_t, a^2_t) + w\cdot \mathcal{P}^{\text{interactive}}(s_{t+1}|s_t, a^1_t, a^2_t).
> $$
> Since the interactive part of the transition can have any weight $w \in (0, 1)$ in such an average, the Lipschitz coefficients can be very small.

---

> > ### Author Response · Authors · 2020-11-22
> > **Review Response (Cont.)**
> >
> > P7, Assumption 4.4: The discussion on the estimation of the marginalized action-value function and the policy update is in Appendix B.
> > --As shown in some recent advances in theories of RL with function approximations [5,6,7], the desired accuracy of MSE can be achieved if both the policies and the marginalized action-value functions are parameterized by overparameterized neural-networks. Here we have the order of $1/t^2$ requirement on estimation error at iteration $t$(which is not the convergence rate of the estimation algorithm itself). As a side note, the approximation of action-value function via neural networks works the same no matter the MDP is entropy regularized or not, since in both cases there are valid (regularized) Bellman equations.
> >
> > --Due to the concentrability assumption (Assumption 4.2), the case where $\nu_t$ is singular with respect to $\nu^*$ in not possible. Using concentrability to enable distribution shifting has long been a quite standard approach in batch RL theories (please see, e.g., [3]).
> >
> > P13, Appendix A/P8, Theorem 4.5/P16, Eq(C.10)-(C.11)/P23, Eq(G.2)&End of Proof of Lemma C.4: Thanks for the valuable questions and advices, we have revised accordingly.
> >
> > [1] Hofbauer, Josef, and William H. Sandholm. "On the global convergence of stochastic fictitious play." Econometrica 70.6 (2002): 2265-2294.
> >
> > [2] Swenson, Brian, and H. Vincent Poor. "Smooth Fictitious Play in N× 2 Potential Games." 2019 53rd Asilomar Conference on Signals, Systems, and Computers. IEEE, 2019.
> >
> > [3] Chen, Jinglin, and Nan Jiang. "Information-theoretic considerations in batch reinforcement learning." arXiv preprint arXiv:1905.00360 (2019).
> >
> > [4] Farahmand, Amir-massoud, et al. "Regularized policy iteration with nonparametric function spaces." The Journal of Machine Learning Research 17.1 (2016): 4809-4874.
> >
> > [5] Liu, Boyi, et al. "Neural trust region/proximal policy optimization attains globally optimal policy." Advances in Neural Information Processing Systems. 2019.
> >
> > [6] Xu, Pan, and Quanquan Gu. "A finite-time analysis of Q-learning with neural network function approximation." arXiv preprint arXiv:1912.04511 (2019).
> >
> > [7] Cai, Qi, et al. "Neural temporal-difference learning converges to global optima." Advances in Neural Information Processing Systems. 2019.
> >
> > [8] Munos, Rémi, and Csaba Szepesvári. "Finite-time bounds for fitted value iteration." Journal of Machine Learning Research 9.May (2008): 815-857.

---

### Official Review · AnonReviewer4 · 2020-10-30
**A nice result on a hard problem**

**Rating:** 8
**Confidence:** 3

**Review:**

This paper studies the problem of learning to play a Nash equilibrium in two-player, zero-sum Markov games.  This is a longstanding problem, with many algorithms proposed but relatively few theoretical convergence guarantees, and most of those either for quite restricted settings or with strong assumptions.  This is in stark contrast to the stateless setting of Normal Form Games, where we have many strong theoretical convergence guarantees.  The main algorithm is a version of the classic fictitious play algorithm.  Like prior adaptations of fictitious play to Markov Games, it operates on the Q-values, but a key novelty (at least in the stateful setting; similar ideas were recently applied in a special case of normal form games by Swenson and Poor 2019) is the use of a particular form of regularization in the best response process.  The main result is that as long as the game satisfies Lipschitz and Concentratability properties for each player when the other plays optimally and the policy updates are sufficiently accurate then play converges to a Nash equilibrium.

I like this paper quite a bit.  It tackles a hard problem  and makes solid progress.  I think the algorithm and analysis are both nice contributions and definitely intend to study the latter further as I think aspects of it may be useful in other settings.  Overall the presentation, while dense, is clear.  However, I believe there are a few issues that could use additional discussion:

 1) Why does the uniqueness, or lack thereof, of the Nash equilibrium not matter to the results?  Quite a bit of prior work has had caveats when they are not unique.  The results seem to hold if the assumptions are true for at least one equilibrium, presumably because of the minimax properties in a zero-sum setting, but I’m not quite clear how this interacts with the assumptions.  For example, if one but not all the equilibra cause the game to satisfy Assumption 4.2 and 4.3 what causes the guarantees to still hold even if initially play gravitates toward some equilibrium where they do not?

2) I’m not quite clear how to interpret the convergence guarantee in Theorem 4.5.  The text after the theorem talks about the policy sequence converging to a neighborhood, while the theorem itself is about the averages across the sequence of policies.  It would help to have some more detailed discussion of exactly what sort of convergence behavior we should expect.

3) I’m intrigued by the observation at the end that this algorithm is Hannan consistent under stronger assumptions.  There has been some work recent work exploring connections between regret minimization and RL and it would be worth discussing a bit how this observation relates to that literature, e.g.:

@inproceedings{hennes2020neural,
  title={Neural Replicator Dynamics: Multiagent Learning via Hedging Policy Gradients},
  author={Hennes, Daniel and Morrill, Dustin and Omidshafiei, Shayegan and Munos, R{\'e}mi and Perolat, Julien and Lanctot, Marc and Gruslys, Audrunas and Lespiau, Jean-Baptiste and Parmas, Paavo and Du{\`e}{\~n}ez-Guzm{\'a}n, Edgar and others},
  booktitle={Proceedings of the 19th International Conference on Autonomous Agents and MultiAgent Systems},
  pages={492--501},
  year={2020}
}

---

> ### Author Response · Authors · 2020-11-22
> **Review Response**
>
> Thank you for your valuable feedback.
>
> 1. Uniqueness of Nash equilibrium: Our smooth FSP involves entropy regularization, which brings curvature to the optimization landscape and thus ensures the uniqueness of the optimal policy. Such an optimal policy lies in a neighborhood of a Nash equilibrium that satisfies Assumption 4.2-4.3.
>
> 2. Assumption 4.2 & 4.3: They are both essential for the convergence guarantee.
>
> --Assumption 4.2 assumes concentrability, which can be satisfied if the game environment is sufficiently explored. Technically, such an assumption allows for changes of measures to the equilibrial stationary distribution $\nu^*$. Intuitively, without sufficient exploration to make this assumption hold, the agents are unlikely to learn the full equilibrium since there is not enough information on the less played states.
>
> --Assumption 4.3 assumes Lipschitz continuity of the Markov game's transition and rewards, which is necessary for the tractability non-stationarity caused by the simultaneous policy updates as well as unavailability of the opponent's Nash equilibrium. Such an assumption enables the entropy regularization to stabilize the non-stationarity. On the other hand, without this assumption, one would expect algorithms (not just our algorithm) fail to converge since the change that the opponent does to the environment is too unpredictable.
>
> 3. Convergence result: The convergence result is delivered in the form of average gap defined using the performance function $\mathcal{J}$ in Eq(2.3). Such a convergence guarantee can be translated to the best policy among the policies generated by the algorithm converges to a neighborhood of a Nash equilibrium. The reason why there is no last iteration convergence established is due to the non-convexity of the performance function. Moreover, the convergence guarantee also depends on how small the regularization parameter $\lambda_i$ is. If $\lambda_i$ is sufficiently small, the algorithm converges to a small neighborhood of a Nash equilibrium.
>
> 4. Hannan consistency: We added more discussion to this property in the revised introduction. While previous works focus on normal-form/extensive form games, our results is under Markov game setting and serve as a first step toward connecting regret minimization with RL.

---

### Official Review · AnonReviewer2 · 2020-11-04
**Important topic. Missing key related work. The weight carried by assumptions can be discussed more.**

**Rating:** 5
**Confidence:** 2

**Review:**

This paper is about the design and analysis of policy optimization algorithms that provably converge to a Nash equilibrium at a sublinear rate for a class of zero sum Markov games, which are one of the simplest settings of multi-agent RL --- in particular, for zero sum Markov games satisfying a Lipschitz regularity condition. Each players adopts an entropy-regularized policy optimization method (which the authors call as smooth Fictitious Self Play).

This is an important topic and the question studied is an important step to take given that we don't know whether Fictitious Self Play is guaranteed to converge in Markov games. However, I am quite surprised by very important related work missing in the paper. For instance, the NeurIPS 2019 paper on "Policy Optimization Provably Converges to Nash Equilibrium in Linear Quadratic Games" is not cited even though it is quite close to the topic of this paper: it also studies a policy optimization algorithm, linear quadratic games are also zero-sum Markov games, the objective studied is also non-convex non-concave. Of course LQ games are special class of zero-sum Markov games, but this paper makes some assumptions like Lipschitz regularity as well. Therefore claims like this paper is the first to prove convergence guarantees of policy optimization algorithms for zero sum Markov games are not quite true. The somewhat less related, but still quite close NeurIPS 2019 paper "Model-based multi-agent RL in zero-sum Markov games with near optimal sample-complexity" is not cited as well. These papers are not obscure: a simple search for the submitted paper's title brings up these papers.

Also, the Lipschitz regularity assumption being made is important enough that it is good to add it in the abstract, as the abstract feels misleading otherwise. And the importance/restrictiveness of this assumption is ideally discussed in more detail in the introduction.

Overall I think this would make a good paper after fixing the above, but not right now.

---

> ### Author Response · Authors · 2020-11-22
> **Review Response**
>
> Thank you for your valuable feedbacks.
>
> 1. References: Thanks for listing the previous works. In our revised version, we've added them to our reference and included more discussions. Although LQ games have some common flavor with the class of games we focus on, there are some substantial differences between our work and the line of works on LQG:
>
> -- Lipschitz assumption: While we assume that the game transition and reward to be Lipschitz, the literature in LQ games states that the best response is Lipschitz with respect to the opponent's policy (i.e., $K^*(L)$ is Lipschitz w.r.t $L$ in [1]).
>
> -- Algorithms are different: [1] uses a double-loop algorithm where the best response to the opponent's current policy is solved at each iteration.
>
> The model-based MARL literature [2] is more different compared to our work. They consider model based case, where the algorithm is essentially planning under a known model. In addition, they study the algorithm's complexity without function approximation.
>
>
> 2. Lipschitz assumption in abstract: We put more emphasis on this assumption in our revised abstract.
>
>
> [1] Zhang, Kaiqing, Zhuoran Yang, and Tamer Basar. "Policy optimization provably converges to Nash equilibria in zero-sum linear quadratic games." Advances in Neural Information Processing Systems. 2019.
>
> [2] Zhang, Kaiqing, et al. "Model-based multi-agent rl in zero-sum markov games with near-optimal sample complexity." Advances in Neural Information Processing Systems 33 (2020).

---

> ### Comment · ~Kaiqing_Zhang3 · 2021-01-16
> **On Related Work**
>
> I was sent the link to this review, in the phase when the public cannot leave comments. I will have to leave some comments now, in favor of the authors.
>
> I am actually the author of the two pieces of related works mentioned by the reviewer. First, I would like to thank the reviewer very much for bringing up our papers. I do agree that the first piece of work on policy optimization for LQ zero-sum games can be relevant. However, I am not sure if the second one is relevant to the topic studied that much. There have been quite a few works on "learning in zero-sum tabular Markov games". I am not very sure about the reviewer's intention of mentioning these two "specific" works. I think to make a good review, it might be better to focus on the techniques/theory/assumptions/writing of the work per se, instead of judging the paper by "mostly" arguing the sufficiency of literature review. It was unfortunate that I was not able to leave these comments when I was notified about this, though I like the paper. I sincerely hope the reviewer can review a paper by reading it in detail, and leaving more constructive comments. It might not be good to simply criticize the lack of related works, and only mention two papers from the same author/group of people, no matter what the intentions of doing so might be.

---

### Author Response · Authors · 2020-11-22
**General Response**


We appreciate all the reviewers for their valuable feedbacks.

We'd like to highlight our contribution, address some common concerns raised by the reviewers and list the changes made to our revised submission.

Contribution

We propose an FSP-type algorithm, which we call smooth FSP, for two-player zero-sum Markov games with Lipschitz continuous transition and reward, and provide convergence guarantee to a neighborhood of a Nash equilibrium under some reasonable assumptions. This serves as a first step towards understanding FSP-type algorithms in multi-agent competitive Markov games. The Hannan consistency further ensured by stronger assumptions also builds a connection between regret minimization in normal-form/extensive-form games and FSP in Markov games. The techniques used in the analysis may also be of independent interest to the analysis of algorithms for more general Markov game settings.

Common Concern

1. Assumption 4.2 (Concentrability): The notion of concentrability is commonly used in literature on RL theory. It was introduced in literature on value iteration theories (see, e.g., [1], where examples can be found) and was later brought to the analysis of policy optimization (see, e.g., [2]). Such an assumption allows for distribution shift from sample distribution to other distributions in analysis, which keeps action-value function evaluations and policy improvements from being meaningless due to information-theoretic bottleneck [3]. While we make such kind of assumption over the policy sequence generated by the proposed smooth FSP, a majority of the existing works make the assumption over arbitrary distributions.

2. Assumption 4.4 (Estimation Error): There are some recent advances in analyzing the error bound of using neural networks parameterization in RL (see, e.g., [4,5]). Although our analysis is carried out on entropy regularized Markov games, the structure of (regularized) Bellman equation remains valid. Thus, the existing error bounds on neural RL are applicable to our estimation error and makes Assumption 4.4 reasonable.

3. Convergence: As can be seen in our analysis, our algorithm converges to an entropy regularized Nash equilibrium, which lies in a neighborhood of a Nash equilibrium that satisfies Assumptions 4.2-4.4. Thus, as long as the two players have sufficiently small (or sufficiently imbalanced) influences to the game transition and reward, the regularization parameters $\lambda_i,i = 1, 2$ can be very small, which in turn leads to a small neighborhood of a Nash equilibrium.

Changes in Revision

1. Discussion: We added more discussion to related works (e.g., linear-quadratic games, variance reduced policy optimization). We rephazed the part discussing the relationship of Assumption 4.3 and the influence introduced in [6]. Also, we made a clearer statement about our Hannan consistency result as a connection and extension to the existing literature on regret minimization of normal-form/extensive-form games.

2. Analysis/Theorem: In the discussions under Theorem 4.5 and in Appendix I, we made a clearer discussion on the conditions that the conditions the Lipschitz coefficients need to satisfy. Also, we made some corrections to the details in our proof (thanks to reviewer 1).

We very look forward to hearing any additional insights or feedback on our responses. Thanks again for the efforts of all the reviewers!

[1] Munos, Rémi, and Csaba Szepesvári. "Finite-time bounds for fitted value iteration." Journal of Machine Learning Research 9.May (2008): 815-857.

[2] Liu, Boyi, et al. "Neural trust region/proximal policy optimization attains globally optimal policy." Advances in Neural Information Processing Systems. 2019.

[3] Chen, Jinglin, and Nan Jiang. "Information-theoretic considerations in batch reinforcement learning." arXiv preprint arXiv:1905.00360 (2019).

[4] Xu, Pan, and Quanquan Gu. "A finite-time analysis of Q-learning with neural network function approximation." arXiv preprint arXiv:1912.04511 (2019).

[5] Cai, Qi, et al. "Neural temporal-difference learning converges to global optima." Advances in Neural Information Processing Systems. 2019.

[6] Radanovic, Goran, et al. "Learning to collaborate in markov decision processes." arXiv preprint arXiv:1901.08029 (2019).

---

### Decision · Program_Chairs · 2021-01-07
**Final Decision**

**Decision:**

Reject

**Comment:**

The paper shows that a form of Fictitious Self-Play converges to the Nash equilibria in Markov games. Understanding the theoretical properties of Fictitious Self-Play is important, however the paper in its current form is not ready for publication. The paper needs a more thorough discussion on related works, the assumptions made, and as pointed out by Reviewer3, the convergence argument needs to be expanded and explained in more detail. Further, I encourage authors to add experiments and compare their algorithm with other methods.